# Enhancing Electrocardiogram (ECG) Analysis of Implantable Cardiac Monitor Data: An Efficient Pipeline for Multi-Label Classification

Amnon Bleich [1,*] , Antje Linnemann [2] , Benjamin Jaidi [2], Björn H. Diem [2] and Tim O. F. Conrad [1]

1   Visual and Data-Centric Computing Department, Zuse Institute Berlin, Takustraße 7, 14195 Berlin, Germany; conrad@zib.de
2   BIOTRONIK SE & Co., KG, Woermannkehre 1, 12359 Berlin, Germany; antje.linnemann@biotronik.com (A.L.); bjoern.diem@biotronik.com (B.H.D.)
*   Correspondence: bleich@zib.de

**Abstract:** Implantable Cardiac Monitor (ICM) devices are demonstrating, as of today, the fastest-growing market for implantable cardiac devices. As such, they are becoming increasingly common in patients for measuring heart electrical activity. ICMs constantly monitor and record a patient's heart rhythm, and when triggered, send it to a secure server where health care professionals (HCPs) can review it. These devices employ a relatively simplistic rule-based algorithm (due to energy consumption constraints) to make alerts for abnormal heart rhythms. This algorithm is usually parameterized to an over-sensitive mode in order to not miss a case (resulting in a relatively high false-positive rate), and this, combined with the device's nature of constantly monitoring the heart rhythm and its growing popularity, results in HCPs having to analyze and diagnose an increasingly growing number of data. In order to reduce the load on the latter, automated methods for ECG analysis are nowadays becoming a great tool to assist HCPs in their analysis. While state-of-the-art algorithms are data-driven rather than rule-based, training data for ICMs often consist of specific characteristics that make their analysis unique and particularly challenging. This study presents the challenges and solutions in automatically analyzing ICM data and introduces a method for its classification that outperforms existing methods on such data. It carries this out by combining high-frequency noise detection (which often occurs in ICM data) with a semi-supervised learning pipeline that allows for the re-labeling of training episodes and by using segmentation and dimension-reduction techniques that are robust to morphology variations of the sECG signal (which are typical to ICM data). As a result, it performs better than state-of-the-art techniques on such data with, e.g., an F1 score of 0.51 vs. 0.38 of our baseline state-of-the-art technique in correctly calling atrial fibrillation in ICM data. As such, it could be used in numerous ways, such as aiding HCPs in the analysis of ECGs originating from ICMs by, e.g., suggesting a rhythm type.

**Keywords:** ECG; ICM; classification; semi-supervised-learning

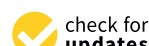



## 1. Introduction

Automated classification (and diagnostics) of ECG data has been a field of active research over the past years, and various methods have been proposed to solve this task (see, e.g., [1–5]). However, most of these approaches require that not only the ECG data themselves have very good quality but also that the data labels for each episode are available and correct. These requirements seem to be fulfilled in most clinical research setups (with professional ECG devices and trained staff) but are rarely met in the world of implantable devices, such as with Implantable Cardiac Monitors (ICMs).

ICMs are an important part of implantable cardiac devices and, as such, play a large role in medical diagnostics. They are becoming more and more common as a diagnostic tool for patients with heart irregularities. These devices are characterized by a relatively simple

classification algorithm that allows the detection and sending of suspicious ECG episodes to HCPs for further inspection. Because this algorithm prioritizes sensitivity over specificity (as it cannot afford to miss heart rhythm irregularities), many of the detections are false positives, i.e., the recorded episodes are often not a real indication of a problem. The outcome is an increasing number of ECG data that are sent regularly to HCPs for manual diagnoses. As a result, the current bottleneck of the ICM in treating patients is not in the device itself, but rather in the scalability in terms of manual analysis performed by HCPs. In other words, due to the rapid growth in the number of sent episodes, it is expected that in the near future, HCPs will not be able to give the proper attention needed for analyzing them [6]. Therefore, to reduce the load on HCPs that analyze episodes recorded by ICMs, an automated method to assist HCPs in annotating and/or suggesting probable arrhythmia in ICM episodes is becoming essential.

Furthermore, most available methods and existing studies for ECG analysis focus on the relatively high quality of at least two-lead ECG signals and contain high-quality, manually assigned labels. In contrast to that, ICMs produce only one-(variant)-lead ECG data—or, more correctly, one subcutaneous ECG (sECG) datapoint—with variant morphology due to variable implantation sites and are of comparably low resolution (128 Hz). In addition, manually labeled data sets often only provide inaccurate labels. Thus, available analysis methods for the better-quality data (two-or-more leads) are hardly suitable for the analysis of the lower-quality ICM data (see also Section 6 for a performance evaluation of these methods on ICM data).

In this paper, we address the above-mentioned problems and present a novel method that allows de novo labeling and re-labeling for ECG episodes acquired from ICM devices. More specifically, the method allows assigning labels to segments of ECG episodes. Moreover, the method is applicable to low-resolution data sets with small sample size, class imbalance, and inaccurate and missing labels. We compare our method to two other commonly used methods on such (or similar enough) data and show the superiority of our method (especially on minority classes). Although the overall performance has room for improvement, it can nevertheless be seen as a milestone on the way to automatically classify such data or e.g., serve as an arrhythmia recommender for HCPs.

It should be noted, however, that our main focus in this paper is on addressing the difficulties associated with data obtained from (ICMs), an area that has received limited attention in previous studies on ECG classification. Understanding the unique characteristics of ICM data is crucial for designing effective solutions, as these data differ significantly from traditional high-quality ECG signals. The insights gained from our analysis of ICM data serve as the foundation for our novel approach, which is tailored to address these specific challenges.

Thus, our main objective is to create an all-encompassing approach that addresses the inherent constraints associated with ICM data, including low resolution, single-lead recordings, class imbalance, and inaccurate labels. Further objectives are (i) to outline the challenges imposed by the unique characteristics of data provided by ICM devices and their importance in the learning and treatment of heart diseases; (ii) to present a pipeline for the analysis of ICM data that outperforms state-of-the-art methods in classifying them; and (iii) to introduce the underlying methods that are our pipeline's building blocks for analyzing ECG data.

To reach these objectives, this paper proposes a novel data analysis pipeline for ICM data. The main highlights of this pipeline area as follows.

1. The novel method allows for the de novo labeling and re-labeling of ECG episodes acquired from Implantable Cardiac Monitors (ICMs). The main goal is to reduce the load on HCPs by assisting in annotating and suggesting probable arrhythmia in ICM episodes.
2. The method is applicable to low-resolution data sets with small sample size, class imbalance, and inaccurate and missing labels. This is a significant contribution as most available methods and existing studies for ECG analysis focus on high-quality

manually assigned labels for at least two-lead ECG signals, which are not suitable for the analysis of low-quality ICM data.

3. The results of our experiments suggest that our new approach is superior to two other commonly used methods on ICM data, especially on minority classes. Although the overall performance has room for improvement, it can be seen as a milestone on the way to assisting HCPs as an arrhythmia recommender.

Before we dive into the details of our new method (Section 4), we will first provide background information about ICM devices and the acquired data in the next section. We will then present our experiments and results (Sections 5 and 6), including a comparison and evaluation with other established methods. Finally, we discuss and conclude our findings (Sections 7 and 8).

## 2. Implantable Cardiac Monitors

Implantable Cardiac Monitors (ICMs) continuously monitor the heart rhythm (see [7] for a detailed overview). The device, also known as an implantable loop recorder (ILR), continuously records a patient's subcutaneous electrocardiogram (sECG). Once triggered, it stores the preceding 50 s of the recording up to the triggering event, plus 10 s after the triggering event. (Note that there are cases where the recorded signal is less than 60 s due to compression artifacts.) The result is an sECG that is (up to) 60 s long and in which we expect to see the onset and/or offset of an arrhythmia (i.e., the transition from an normal sECG to an abnormal one or vice versa). The stored episodes are sent to a secure remote server once per day.

Possible detection types include atrial fibrillation (AF), high ventricular rate (HVR), asystole, bradycardia, and sudden rate drop.

Detections (or triggers) are based on simple rules and thresholds, namely (i) the variability of the R-R interval exceeding a certain threshold for a set amount of time for AF, (ii) a predefined number of beats exceeding a certain threshold of beats-per-minute for HVR, (iii) a mean heart rate below a certain threshold for a set time for bradycardia, and (iv) a pause lasting more than a set amount of time for asystole.

In summary, data are retrieved daily from the device through a wireless receiver for long-distance telemetry. The receiver forwards the data to a unique service center by connecting to the GSM (Global System for Mobile Communication) network. The Service Center decodes, analyzes and provides data on a secure website, with a complete overview for the attending hospital staff. Remote daily transmissions include daily recordings of detected arrhythmia. Transmitted alerts were reviewed on all working days by an HCP who is trained in cardiac implantable electronic devices.

### 2.1. Challenges with ICM Data

Data acquired from an ICM device have several quite unique characteristics compared to data acquired using more commonly used ECG devices, e.g., in a clinical setting. Some of the key challenges include the following.

**Variant ECG morphologies**: Unlike e.g., the classical Holter ECG recording device, the ICM device allows for a variant implant site relative to the heart, which results in a variant ECG morphology. This variance in ECG morphology introduces an artifact that makes it difficult for machine learning models to compensate for.

**No "normal" control group**: Due to the nature of ICMs, a record is only sent when an abnormal event occurs. This means that the available data consist of—almost only—ECG episodes that contain an unusual event—which triggered the sending of the episode.

**Inaccurate labels**: ECG episode analysis and diagnosis require special attention from HCPs. However, since the availability of such professionals is limited and the number of episodes that require labeling is large, the time dedicated to manually labeling an episode is minimal and results in a degree of inaccuracy in the labels caused by a human factor. Furthermore, another degree of label inaccuracy is added as the different heart rhythm definitions are not completely consistent between HCPs, and thus, different labels can be

given via different HCPs to similar heart rhythms. This is further increased by the fact that many annotation platforms allow the assignment of only global labels. This means that an annotation does not have a start and end time but rather refers to the entire episode. This is unsuitable for the typical data set produced by ICMs, as these consist in the majority of cases of 60 s long episodes, in which the onset of a particular arrhythmia is expected—i.e., by definition, a single heart rhythm (viz. label) cannot relate to an entire episode. Since the accuracy of a supervised model is bound by the accuracy of the labels and inaccuracy in the labels reduces its learning efficiency, such an issue imposes a challenge in learning ICM data.

**Class imbalance**: The false detection ratio of the device results in the vast majority of episodes being Sinus episodes with some artifact that triggered the storing of the episode.

**Small multi-label training-sets**: Most supervised learning models of ECG data require relatively large data sets of manually annotated ECG episodes. However, since it involves highly proficient personnel to thoroughly review every sample in the training data set (especially in the case of multi-class labeling), these tend to be relatively small.

**Low resolution**: ICMs compress data before storing them, resulting in a low sampling rate (128 Hz compared to a minimum of 300 Hz in comparable applications).

**Noise**: Movements of patients might result in noisy amplitudes, which could trigger a device recording—this could result in the irregular class distribution in the data set, especially regarding noise.

## 3. Related Work

Various methods have been published in recent years for the automatic diagnosis of ECG data (e.g., [1–3]) or other feature-based heart data (e.g., [8]). In [9], Stracina and co-workers give a good overview of the developments and possibilities in ECG recordings that have been made since the first successful recording of the electrical activity of the human heart in 1887. This review also includes recent developments based on deep learning. Most of the available analysis methods aim at the classification of ECG episodes, e.g., to determine whether an ECG signal indicates atrial fibrillation (AF) or any other non-normal heart rhythm.

Most state-of-the-art solutions for ECG classification use either feature-based models such as bagged/boosted trees [10–12], SVM, or k-nearest neighbor [9] or deep learning models such as convolutional neural networks (CNN) [13]. While deep learning approaches usually use the raw 1d ECG signal [14–17], feature-based ECG arrhythmia classification models involve the extraction of various features from ECG signals, such as time-domain and frequency-domain features.

Spectral features are derived from the frequency components of the ECG signal and are extracted, e.g., using Wavelet transform, Wavelet decomposition, and power spectral density analysis [18].

Time-domain features, on the other hand, are based on the P-waves, QRS complexes, and T-waves, including their regularity, frequency, peaks, and onsets/offsets [9]. Wesselius and co-workers [19] have evaluated important time-domain features, such as Poincaré plots or turning point ratios (TPRs), as well as other ECG wave detection algorithms (such as P- and F-waves) and spectral-domain features, including Wavelet analysis, phase space analysis, and Lyapunov exponents. These time-domain features usually require the detection of the respective waves, with the majority of algorithms focusing on the detection of QRS complexes due to their importance in arrhythmia classification and the fact that they are relatively easy to detect. For instance, a few of the most important features for ECG signal classification are based on RR intervals, including the time between two consecutive R peaks, with R-peaks commonly detected via the detection of QRS complexes [20]. Popular QRS detectors that are used by, e.g., [21] are *jqrs* ([22,23]), which consists of a window-based peak energy detector; *gqrs* [24], which consists of a QRS matched filter with a custom-built set of heuristics (such as search back); and *wqrs* [25], which consists of a low-pass filter, a non-linearly scaled curve length transformation, and specifically designed decision rules.

*3.1. PhysioNet/CinC 2017 Challenge*

We were also interested in methods that have been shown to be successful in applied settings, such as challenges or real-world comparisons. For this purpose, we searched the literature for recent surveys. We used PubMed in January 2023 with the query "single lead ECG machine learning review" OR "single lead ecg AI review" and restricted the time frame to 2020 and later. The results were manually inspected and led us to a challenge that took place in 2017 and was related to our approach.

The *PhysioNet/Computing in Cardiology 2017 Challenge* [26] was a competition organized by the PhysioNet initiative—a research resource that offers open access to databases of physiological signals and time series, and the Computing in Cardiology (CinC)—an annual conference that focuses on the application of computer science techniques to problems in cardiovascular medicine and physiology. In 2017, the two collaborated to conduct the PhysioNet/CinC 2017 Challenge, which was focused on the detection of atrial fibrillation (AF) from short ECG recordings. This challenge was intended to encourage the development of novel algorithms and techniques and attracted experts from diverse fields, including computer science, cardiology, and signal processing, thereby facilitating a multidisciplinary approach to the problem. The results of the competition demonstrated the potential of data-centered techniques, such as deep learning and ensemble methods, for enhancing the accuracy of AF detection.

The best-performing approaches include solutions based on deep learning and decision trees. For example, the winning solution of the challenge by Datta and co-workers [4] approaches multi-label classification via cascaded binary Adaboost [12] classifiers along with several pre-processing steps, including noise detection via signal filters, PQRST point detection, and 150 features that are derived from them. Another example is the approach by one of the top-scoring teams, which proposes a deep-learning-based solution [13], consisting of a 24-layer CNN and a convolutional recurrent neural network.

Two of the top-scoring algorithms of this challenge, for which source code was publicly available, are (1) a feature-based approach that extracts 169 features for each episode and uses an ensemble of bagged trees, as well as a multilayer perceptron for classification, and (2) a deep learning-based method using a 34-layer ResNet. Both approaches have been compared in a paper by Andreotti and co-workers [27,28] and are still considered state-of-the-art approaches for short single-lead ECG data classification. We will use these two approaches as a baseline to compare to our approach (see Section 5.1 for more details).

## 4. Our Methodological Details

This section provides a detailed description of our pipeline, which we developed to analyze and classify 60 s episodes of single-lead subcutaneous ECG (sECG) data. This pipeline is intended to be run on a server (as opposed to on the ICM device), and therefore, it can be more computationally intensive than the algorithm run on the device. The main steps include segmentation, noise detection, embedding, and clustering (see Figure 1). The resulting clustering is then used to construct the final classifier, which allows for the prediction of labels for new episodes. Each of these pipeline steps will be discussed in detail in the following subsections.

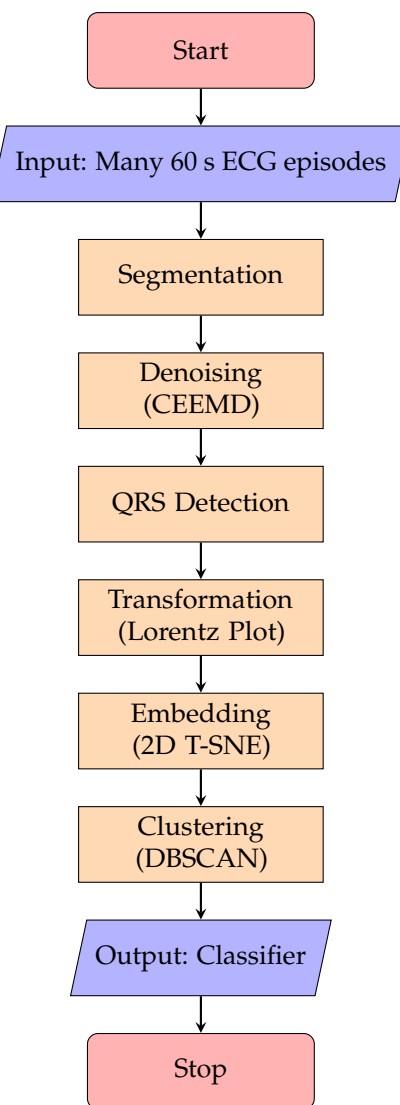

**Figure 1.** Overview of the method's main building blocks.

### 4.1. Input Data

The input to the pipeline is a data set consisting of multiple 60 s sECG episodes for which at least one label has been assigned by medical experts. Every episode potentially contains the onset and/or offset of arrhythmia (see Section 2) and thus, by definition, more than one rhythm type per episode. The annotation platform used for our data set allows assigning labels to episodes in a manner of selection out of a bank of 38 possible labels. However, due to the relatively small data set, in order to reduce the number of classes and thus increase the number of episodes belonging to each class, the various labels are grouped into 5 categories: normal, pause, tachycardia, atrial fibrillation (denote *aFib*) and noise.

### 4.2. Training

The training pipeline consists of several steps, which are shown in Figure 1. The following paragraphs explain the building blocks of this pipeline in more detail.

#### 4.2.1. Segmentation

In order to have homogeneous episodes, i.e., one heart rhythm per episode (which correlates to one label per episode), we divide the 60 s episodes into six 10-s, non-overlapping sub-episodes. Each sub-episode is assigned the label of its parent episode. Note that this label is presumed to be inaccurate, as it could have been chosen due to a rhythm that

appears in another sub-episode of that episode, and the later steps in the pipeline will correct for this.

### 4.2.2. Noise Detection

ECG data—and in particular data from ICM devices—typically contain noise, such as baseline wander, high/low-frequency interruptions, and the like ([29]). Although certain technologies in ICM are aimed at reducing this effect ([30]), filtering out noise is essential and improves downstream processes, such as the detection of the QRS complex. Several approaches have been adjusted from the general field of signal processing, such as the extended Kalman filter, median, low- or high-pass filters, or Wavelet transform [31–34]. In the proposed method, we suggest using an algorithm based on empirical mode decomposition (EMD) [35,36], which has been shown to deal well with high-frequency noise and baseline wander. The pseudocode in Algorithm 1 describes a step-by-step description of the CEEMD (Complete Ensemble Empirical Mode Decomposition [37])-based algorithm used in order to detect noisy ECG signals. It first decomposes the signal into its IMFs (intrinsic mode functions [38]) using CEEMD, where the first IMFs account for the high-frequency signal and the last IMFs account for low frequency (e.g., baseline wander; see Figure 2a). It then sums up the first 3 IMFs (the components accounting for the high-frequency signal, denoted by *Hn*) and checks over a sliding window whether *Hn* in that window contains a value greater than a certain threshold and if it has more than a set number of zero-crossings. The result of this is a binary vector with 1 s where both conditions are met and 0 s otherwise (denoted as *Gn*—see Figure 2b). Finally, if a certain *gate* (a sequence of 1 s in the *Gn* vector) is longer than a set threshold, the corresponding part of the original ECG signal is marked as noise. All the parameters (namely *threshold, window size* and *gate length*) were selected empirically using the Jaccard index to quantify the overlap between the detected noise and the segments annotated as noise in the test data set, as the objective function. To conclude, in order to see the relevance of the noise-detection algorithm to our data, we marked all the noisy segments in it and compared them with the assigned labels. The results of this step can be found in the results section (Section 6).

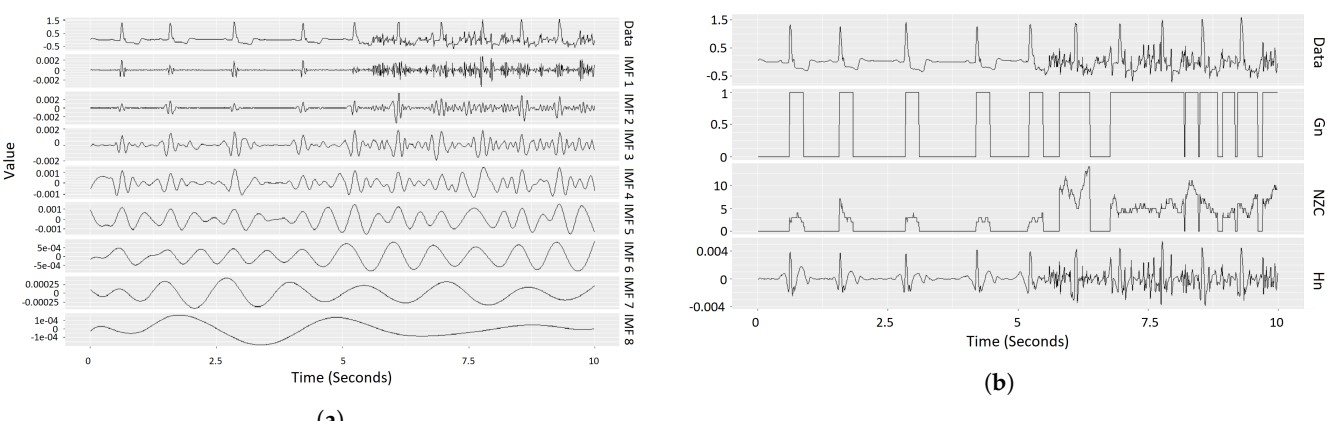

**Figure 2.** Example of our noise-detection algorithm. Left: intrinsic mode functions (IMFs) of an sECG. Right: components derived based on IMFs for actual noise detection. (For details, see description of Algorithm 1.) (**a**) Example for the CEEMD method: The first row shows a single sub-episode (10 s) of an sECG episode. The signals in the rows below show the first 8 intrinsic mode functions (IMFs) resulting from running CEEMD on this sub-episode. The IMFs are different components with decreasing frequencies of the sub-episode, such that the summation of all the IMFs results in the given sECG sub-episode. (**b**) Visualization of the noise-detection step, as described in Algorithm 1. The first row shows the ECG data (same as in the left sub-figure). The rows below (Gn, NZC, Hn) represent the derived components, as explained in Section 4.2.2, where Hn is the sum of the first three IMFs (IMFs 1, 2 and 3 in the left sub-figure), NZC is the number of zero-crossings in Hc within a

given window size ($\sim$0.23 in our case), and Gn is a "gate" vector with 1 s where NZC is strictly greater than 1 ($\geq$2) and 0 otherwise. The component dependencies are hierarchical (displayed in reverse order): Hn computation relies on the IMFs, NZC is dependent on Hn, and Gn is based on NZC. Finally, if a gate in Gn (consecutive Gn = 1 segment) is longer than a set time (in our case, 0.75 s), the corresponding segment in the sECG is marked as noise.

---

**Algorithm 1** Our CEEMD-based noise-detection algorithm

---

1: *ceemd* = CEEMD of ecg wave signal with sift_num = 10; 100 iterations and Gaussian noise
2: Define:
   a: $Hn$ = Sum of IMF1, IMF2 and IMF3 of *ceemd*
   b: *threshold* = 0.85 th quantile of abs(Hn)
   c: *window_size* = 0.234375 s
3: **for** $x$ in each *window_size* long, center-aligned sliding window on *Hn* **do**
4:    **if** $max(abs(x)) > threshold$ **then**
5:       $NZC[window\ start]$ = number of zero crossing in $x$
6:    **else**
7:       $NZC[window\ start] = 0$
8:    **end if**
9: **end for**
10: Pad NZC with leading 0 s to match Hn length
11: **for** $i$ in $1 \dots length(NZC)$ **do**
12:    **if** $NZC[i] > 1$ **then** $Gn[i] = 1$ **else** $Gn[i] = 0$
13: **end for**
14: **for** *gate* in consecutive 1 s series in *Gn* **do**
15:    **if** length of gate $> 0.75$ s **then**
16:       mark gate start/end times as start/end of noise segment
17:    **end if**
18: **end for**

---

### 4.2.3. Embedding

To be able to deal with the high-dimensional ECG data and prevent overfitting (due to the curse-of-dimensionality problem) as much as possible, we perform a projection of the data (approximately 7600 data points, i.e., 7600 dimensions for an episode or 1300 dimensions for a sub-episode) into a low-dimensional (2-dimensional) space. We suggest the following steps for the dimensionality reduction (see Figure 3). First, we (1) perform QRS complex detection on the ECG signals to identify the R-peaks (Figure 3a). Based on the R-peaks, we can (2) transform the ECG signals into a Lorenz plot (Figure 3b), which we (3) discretize into a 2D histogram, using an $N \times N$ grid (Figure 3c). Then, we (4) flatten the 2D histogram into a 1D vector with $N \cdot N$ elements. The last step is (5) to use t-SNE to project the resulting vectors into the 2D space to obtain the final embedding. This is described in more detail as follows.

(1) **R-peak Detection**: The R-peak is a component of the QRS complex that represents the heart in an ECG signal. There are several algorithms for QRS detection, such as gqrs [24], Pan-Tompkins [39], maxima search [40], and matched filtering. In our pipeline, we use an aggregation of these four approaches via a kernel-density-estimation-based voting system.

(2) **Lorenz Plot Transformation**: In the context of ECG, a *Lorenz plot* (also called Poincaré plot) [41], is the plotting of $dRR(i)$ vs. $dRR(i+1)$ for each R peak $i$ in an ECG signal, with $dRR(i) = RR(i+1) - RR(i)$, where $RR(i)$ is the i-th R-R interval (the time between two consecutive R peaks). This visualization is often used by HCPs to detect arrhythmia.

(3) **Discretization**: In order to produce a 2D histogram of the Lorenz plot, the plot is divided into equal-sized 25 bins by overlaying it with a 5-by-5 grid. The histogram

is thus a count of points (viz. dRR(i)/dRR(i+1) intervals) in each bin. The result is a 5-by-5 matrix that is flattened to form a 25-dimensional vector.

(4) **t-SNE Embedding**: t-SNE is an algorithm for dimensionality reduction [42]. It is primarily meant for visualization (i.e., projection of multi-dimensional data on a 2D or 3D plane), and since it does not preserve distance between points, it is not suitable for distance-based clustering (such as DBSCAN—the algorithm used in our work). However, since it maximizes the likelihood that contiguous data points will not be in the same cluster, it performs better than alternative embedding methods, such as LDA [43] and PCA [44].

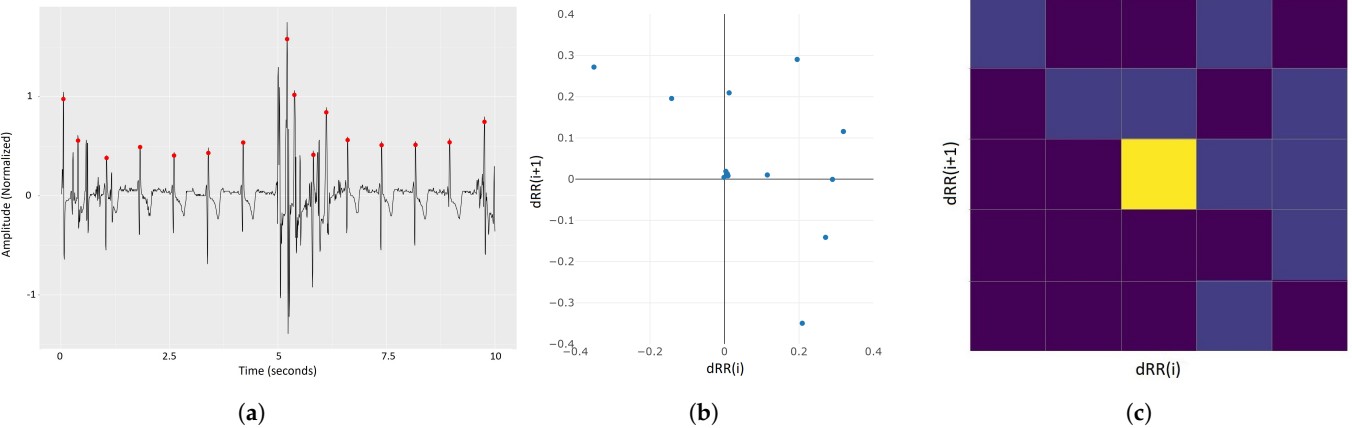

(a)  (b)  (c)

**Figure 3.** ECG embedding steps (excluding the final t-SNE step). (**a**) Raw sECG signal with R-peaks marked with red dots. (**b**) Lorenz plot of the sECG signal in sub-figure (**a**), plotting the relations between consecutive R-R intervals in (**a**). (**c**) Two-dimensional histogram of the Lorenz plot in figure (**b**) with 5 × 5 bins. The colors (values) of the histograms account for the number of points in the Lorenz plot in each bin–brighter colors account for higher values. The latter is flattened to form a 25-dimensional vector, which is given as input to the t-SNE algorithm to provide a 2D representation of the sECG signal.

4.2.4. Clustering

There are various clustering approaches described in the literature, with popular ones including k-means, spectral clustering, or DBSCAN [45]. In our pipeline, we chose the DBSCAN algorithm since it outperformed other approaches we tried on our data sets. DBSCAN stands for Density-Based Spatial Clustering of Applications with Noise, and, as its name suggests, it clusters together areas with high density of points separated by areas with low density while leaving points in sparse areas as outliers.

In order to select the maximal distance parameter, we created a k-distance plot with $k = 2 \times$ the number of features, which in the case of 2D t-SNE is $k = 4$. The reasoning for this is that the distance threshold should be proportional to the number of dimensions to account for the sparsity resulting from it, with $2 \times k$ being a common rule of thumb. Once the minimal point number is set, the *k*-distance plot is a sorted representation of the average distance of each point in the data to its k-nearest neighbors. The area after the elbow would therefore represent points in the data that could be considered outliers in this sense (as finding k nearest neighbors would require searching increasingly further), so this could mark a reasonable maximal distance to look for neighbors. The latter resulted in a distance of eps = 1.5. We then explored the neighborhood of that point interchangeably with the minimal number of points. For a minimal number of points, we tried different values on a scale from a conservative lower bound to an upper bound (2 to 30) until we found visually satisfying results, with the number of outliers (points not belonging to any cluster) not exceeding 10% of the number of sub-episodes. The optimal values for our case were 0.75 for the maximal distance and 15 as the minimal number of points.

### 4.2.5. Building the Classifier

From the clustering results from the previous paragraph, a classifier that enables the label prediction of previously unseen episodes can be created. In order to classify a sub-episode that belongs to a certain cluster (denote cluster $c$), we compute the $p$-value of the label proportions for all labels in that cluster, given the label proportion in the data set. The $p$-value is computed using the cumulative binomial distribution defined as

for cluster $c, \forall l \in$ labels,

$$F(k_{cl}, n_c, p_l) := Pr(X < k_{cl}) = \sum_{i=0}^{k_{cl}-1} \binom{n_c}{i} p_l^i (1 - p_l)^{n-i} \tag{1}$$

i.e., the probability for a maximum of $k_{cl}$ successes in $n_c$ trials with probability $p_l$ for success in a single trial. In our case, $k_{cl}$ denotes the observed number of sub-episodes with label $l$ in cluster $c$, $n_c$ denotes the size of cluster $c$, and $p_l$ denotes the proportion of label $l$ in the data set (defined as $p_l := \frac{d_l}{d}$ with $d_l =$ label count in the data set and $d =$ data set size).

Formula (1) is used to assign $p$-values of $P(l, c) = Pr(X \geq k_{cl}) = 1 - Pr(X < k_{cl})$ to each label $l$ in cluster $c$ and the label with the lowest $p$-value is assigned to the sub-episode. In other words: for a given label and cluster with a label count of $k_{cl}$ in that cluster, the $p$-value computed is the probability of observing at least $k_{cl}$ sub-episodes of that label in that cluster given the label's proportion in the data set and the cluster size. The lower the probability, the higher the significance of that cluster being labeled with that label, and therefore the label with the lowest $p$-value in a cluster is set to be the label of this cluster, and all sub-episodes clustered to it will be labeled as such.

To use the classifier for predicting the labels of new, unseen episodes, the given ECG episode is segmented into 10 sub-episodes and each is projected in the same way as described above. Then, we perform a k-nearest neighbor search (with $k = 1$) on each sub-episode. We assign the cluster of the nearest neighbor of each sub-episode as the cluster of that sub-episode and assign the label accordingly.

Note that several approaches could be taken for the choice of label based on the $p$-value, according to the use case. For example, if one wants to ensure high sensitivity for several classes at the expense of specificity for, say, a pause rhythm, a threshold could be set such that if the $p$-value for pause is lower than that threshold, the cluster's label will be "pause" regardless of other classes with possibly lower $p$-values. This is discussed more in Section 7. On another note, since each sub-episode in the training set is associated with a cluster, and each cluster is associated with a label, the e-labeling of sub-episodes that have a different label than the cluster's label can be carried out, thereby addressing the challenges regarding inaccurate labels mentioned in Section 2.1.

## 5. Experimental Setup

In the following sections, we introduce the two algorithms selected as comparative benchmarks for our novel method and detail the data set used for this study.

### 5.1. Baseline Methods

Based on our literature search (Section 3.1), we selected two algorithms as baseline methods that we use to compare our new method to the following: (1) a feature-based approach that extracts 169 features for each episode and uses an ensemble of bagged trees as well as a multilayer perceptron for classification and (2) a deep-learning-based method using a 34-layer ResNet. The two methods were used as proposed by the authors, with the following adjustments made to match our study.

- The feature-based approach uses Butterworth band-pass filters that were adjusted for lower frequency episodes (as the original solution deals with a frequency of 300 Hz and our data have a frequency of 128 Hz).

- Class imbalance—the original method deals with this problem by adding episodes from minority classes (namely AF and noise) from additional ECG data sets. Since we wanted to evaluate the case where the data are imbalanced, we skipped this step.
- In the deep learning approach, padding is performed for episodes shorter than the selected size, in our case 60 s. Due to the nature of labeling in our test data set and the fact that heart rhythms tend to last a few seconds rather than a minute, the test samples must be significantly shorter than 60 s (10 s in our method) so every sub-episode was padded by 0 values to complete a 60 s episode.

### 5.2. Data

The data set used for this study was produced by the BIOTRONIK SE & Co KG BIOMONITOR III and BIOMONITOR IIIm [46] and anonymized for this analysis. The data produced by the device are characterized by single-lead, 60 s long ECG episodes with a sample rate of 128 Hz (7680 data points per 60 s episode).

Out of the 60 s stored episodes, 3607 were labeled by up to three HCPs, each assigning a global label (a label is given to an entire episode, although more than one label can be given if the HCP recognizes more than one heart rhythm in an episode) and 4710 episodes with similar characteristics but where the labels were assigned a start and end time (i.e., per segment as opposed to a global label). In this study, the former was used for learning (denoted as the *training set*) and the latter for evaluation (denoted as the *test set*).

It should be noted that although the test set labels resolve the primary problem with the training data set—that a label is assigned to the entire episode—they might still have additional inaccuracies, and thus, the test set is still not as trustworthy as ground truth labels.

The labels of the training and test sets are assigned as follows: The training set has global labels (i.e., one or more labels for the entire episode without start/end times—see Section 5.2 for details), and therefore, the labels of all sub-episodes are derived from those of the parent episode, and if more than one label is present, the episode is duplicated. The test set has non-overlapping segmented labels assigned by a different annotation platform that allows for such annotations (i.e., each label has start and end times—see Section 5.2 for details), and so a sub-episode is assigned the label that has a start and end time that overlaps at least half of it (5 or more seconds in case of 10 s segmentation). If there is no such label, the sub-episode is removed from the data set. The label distribution of the data sets can be seen in Figure 4. The figure highlights the class imbalance, where, e.g., in the training set, 2795 episodes represent normal episodes, 546 pause episodes, 467 tachycardia episodes, 985 aFib episodes, and 932 noise episodes. (Note that multiple labels per episode are counted once for each label, and therefore, the number of labels sums up to more than the number of episodes.)

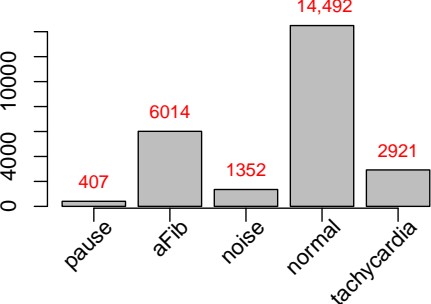

**Figure 4.** Distribution of labels in the training set (**left**) and test sets (**right**). Note the difference in the distributions is due to the way the data were created; see Section 5.2 for details.

The reasoning for the selection of test and training sets is that the training set presents the difficulties referred to in Section 2.1 and the test set contains similar difficulties, but due to the different labeling systems, the labels are assumed to be more accurate.

## 6. Results

This section presents the results of our experiments and compares the performance of our proposed method for classifying cardiac rhythms to the described baseline methods. Our prediction pipeline involves a series of steps that enable the generation of sECG labels, as described above. The following sections describe the application of our pipeline to sECG data and the achieved results, starting with the noise-detection step.

### 6.1. Noise Detection: Only Limited Effects

Our proposed pipeline begins with the segmentation of the given episodes, followed by the noise-detection step. To understand the influence of this noise-detection step, we performed two experiments: (A) We performed the full pipeline without the noise-detection (and -removal) step. (B) We performed the full pipeline, including the noise-detection step. Here, 1444 sub-episodes (out of 25,186) were identified as noise and were removed from further processing. We then looked at the ground-truth labels of the sub-episodes that were labeled as "noise" in the noise-identification step. As shown in Figure 5, the majority of the sub-episodes detected were indeed noisy parts of the ECG data.

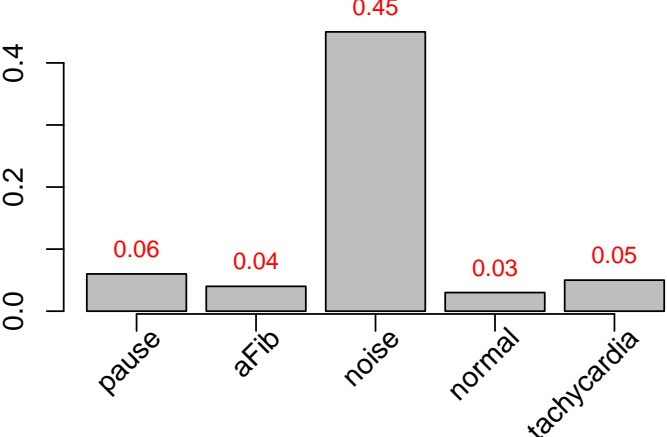

**Figure 5.** Test set CEEMD-based noise detected sub-episodes' label distribution. Ground-truth-label ratio distribution of the 1444 sub-episodes in our test data, detected using our CEEMD-based noise-detection method (values in Figure 1 divided by the number of sub-episodes of each label in the entire data set (e.g., 0.45 for noise means that 45% of the sub-episodes labeled as noise are detected by the algorithm—it should be noted that the algorithm only detects high-frequency noise, and, as such, presumably some of the 55% undetected sub-episodes labeled as noise are of a different kind of noise e.g., saturation/signal loss).

Afterward, we compared the overall performance of pipeline A ("without noise-detection step") and pipeline B ("with noise-detection step"). The comparison reveals that removing the noisy sub-episodes before the training does have only a limited effect overall (see Table 1 for details). The only class with a significant change in performance is the class "noise" with an increased F1-score from 0.24 to 0.30 and, more significantly, an increased sensitivity from 0.34 to 0.61.

**Table 1.** Results of the baseline vs. our method. "Baseline A" and "Baseline B" correspond, respectively, to the feature-based (A) and deep-learning-based (B) methods. "Our A" and "Our B" correspond, respectively, to the method presented here without and with the noise-detection pre-step. All methods were trained on the inaccurately labeled training data set and tested on our segment-annotated data set. The numbers in parentheses correspond, respectively, to specificity and sensitivity (spec/sens). Numbers in bold mark the best result in each row (each ground-truth arrhythmia type).

| | | **Performance—Baseline vs. Our Method—F1 Score (Specificity/Sensitivity)** | | | |
|---|---|---|---|---|---|
| **Class** | **n** | **Baseline A (Feat.-Based)** | **Baseline B (DL-Based)** | **Our A (w/o Noise Det.)** | **Our B (w/ Noise Det.)** |
| Normal | 14,492 | **0.71 (0.35/0.81)** | 0.47 (0.62/0.40) | 0.65 (0.84/0.54) | 0.62 (0.89/0.48) |
| Pause | 407 | 0.03 (1.00/0.02) | 0.03 (0.99/0.03) | **0.10 (0.83/0.61)** | **0.10 (0.84/0.54)** |
| Tachycardia | 2921 | 0.00 (1.00/0.00) | 0.04 (1.00/0.02) | **0.65 (0.96/0.65)** | 0.64 (0.96/0.60) |
| aFib | 6014 | 0.37 (0.86/0.34) | 0.38 (0.80/0.38) | 0.49 (0.85/0.47) | **0.51 (0.85/0.51)** |
| Noise | 1352 | 0.12 (0.94/0.13) | 0.18 (0.66/0.66) | 0.24 (0.91/0.34) | **0.30 (0.86/0.61)** |

*6.2. Embedding*

Applied to our 10 s segmented and combined test and training data set, the pipeline described in Section 6 resulted in a 2D sub-episode projection that is illustrated in Figure 6 with the class distribution of the training sub-episodes highlighted in green. The figure visualizes the effectiveness of the embedding process, where the different classes tend to concentrate in different areas of our 2D space.

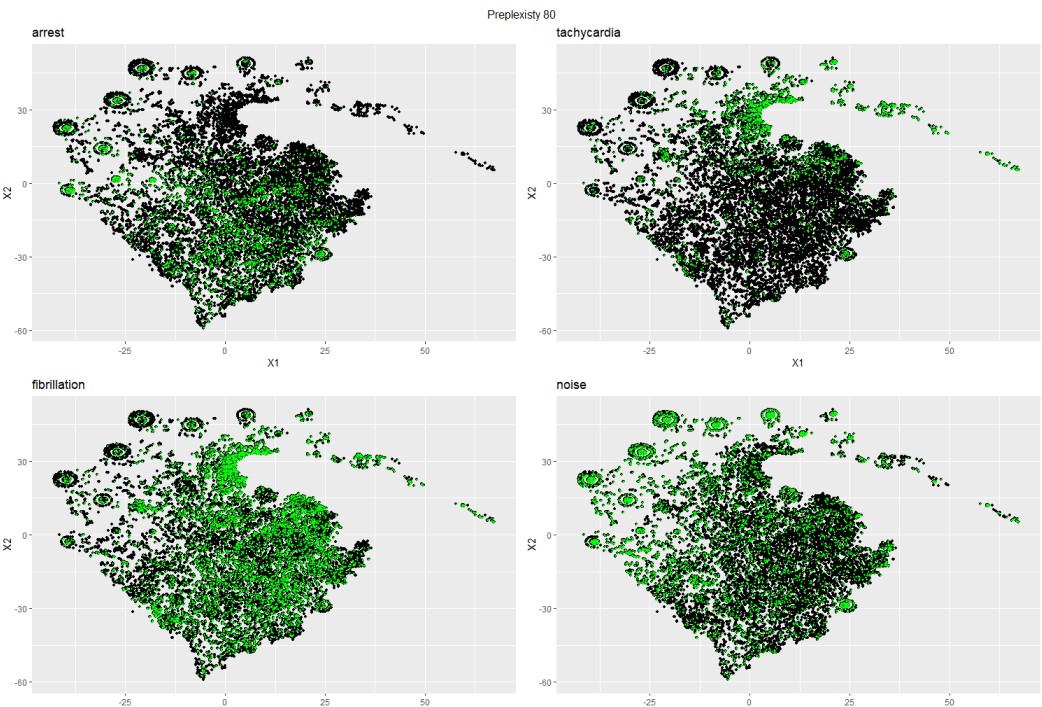

**Figure 6.** Two-dimensional t-SNE projection of the Lorenz plot histogram embedding of all ECG sub-episodes (training and test data). Highlighted in green are sub-episodes from the training set with the respective label for each sub-plot ((**top-left**): Arrest; (**top-right**): Tachycardia, (**bottom-left**): Fibrillation; (**bottom-right**): Noise). Note that every sub-plot is expected to contain green-highlighted sub-episodes outside of the high-density areas because all sub-episodes of an sECG episode in the training set inherit the–potentially wrong–label of its parent episode (see Section 4.2.1 for details).

### 6.3. Clustering

We used the DBSCAN algorithm to cluster the 2D projection of the embedded points from the previous step. Figure 7 shows the results of this clustering step applied to the full data set. One can see that the clustering algorithm manages to separate the data set in an organic manner via areas with a lower density of data points (as opposed to, e.g., arbitrary pentagonal clusters, or many small/few large clusters). In addition, the visualization of the clusters combined with the class distribution depicted in Figure 6 can suggest probable labels that will be given to certain clusters as part of the labeling procedure (for clusters with a high density of a certain class).

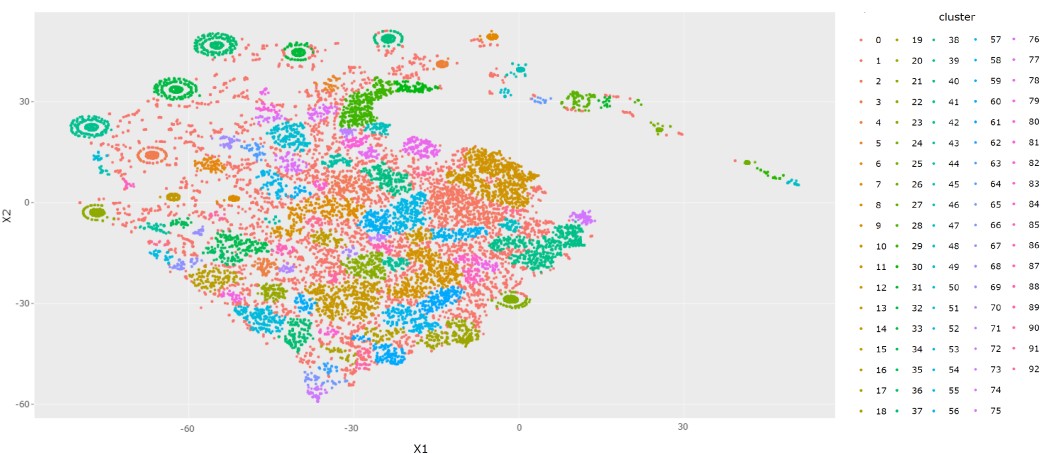

**Figure 7.** t-SNE base 2D visualization of the DBSCAN clustering of all training set and test set data after the embedding of the QRS Lorenz plot 5 × 5 bins histogram (see Section 4.2.3 for more details).

### 6.4. Overall Performance: Better Label Prediction and Lower Runtime

The overall performance of our new method compared to the two selected competitive methods is summarized in Table 1. The results show that our method's prediction outperforms the other two methods in almost all classes, based on the F1 score, especially in the minority classes. As can be seen in the table, there is quite a difference in performance between the various heart rhythm types. The class "normal rhythm" seems to be the easiest to classify. This might be due to the large number of training samples of the "normal" type in the training set —almost half (49.1%) of all samples have this label. This is the only class where the baseline method outperforms our method, by a difference of 0.06 in the F1 score. Moreover, this is arguably the rhythm we are the least concerned about, and even in this rhythm, our method significantly outperforms the baseline in specificity with 0.84 without noise (*Our A*) cleanup and 0.89 with noise cleanup (*Our B*) compared to 0.35 and 0.62 for the feature-based and deep learning methods respectively. In the case of a normal heart rhythm, in most use cases (such as alerting patients of possible heart issues they might have), specificity is preferred over sensitivity—i.e., we are required to correctly reject normal labels for non-normal episodes even at the expense of incorrectly rejecting normal labels.

Furthermore, it is apparent that our method deals better with class imbalance with the performance of minority classes significantly better than the baseline. In addition, our method has a significantly lower runtime: while our method needs about 10 min on the machine used for this study, the baseline methods need several hours to finish computation on the same machine. Also, our method has the ability to "fix" inaccurate training labels as well as label shorter segments of the episodes via semi-supervised learning.

## 7. Discussion

The proposed method aims to reduce clinician workload by recommending and emphasizing certain insights, but not with a focus on making autonomous decisions. The performance of the method supports this tendency, as, while it outperforms existing

methods for the use case for which it is intended, it is not as accurate as a professional's manual diagnosis. In this situation, the use case can affect model parameters, which could have a significant effect on the model's efficacy. According to the authors of this work, one of the strengths of this method is its adaptability to various scenarios through small modifications. In order to decrease false negatives despite an increase in false positives (falsely assigning normal episodes with non-normal labels), one could call the non-normal episode even if it has a higher $p$-value than the normal label. Another option is to set distinct thresholds for various labels, as required by the use case. Such modifications could even be made to a point where the false-negative rate is close to zero, thereby reducing the number of episodes sent to clinicians. In such a case, additional research should be conducted to determine whether its use as a filter on top of the ICM device's algorithm reduces the number of false positives without increasing the number of false negatives. Compared to the Lorenz plot histogram, larger modifications are possible, such as using a 2D projection of a distinct set of characteristics. Such modifications, however, necessitate additional research.

Another noteworthy aspect of the method is its potential to be incorporated into a larger decision-support system, which could combine the strengths of various techniques or algorithms. In such a framework, the proposed method could serve as a pre-processing phase or supplementary tool to improve the system's overall performance. This integration could facilitate stronger decision making and further reduce the workload of clinicians.

When implementing the proposed method in a real-world clinical setting, user-centered design principles should also be considered. Usability, interpretability, and user acceptability will be vital to the successful adoption of the method by healthcare professionals. Future research could investigate methods for enhancing the method's user interface and providing meaningful feedback to clinicians, thereby ensuring that the system's recommendations are in line with their requirements and expectations.

Finally, the method's ethical implications must be exhaustively examined. As the method is intended to reduce clinician burden rather than replace human decision making, it is crucial to ensure that its implementation does not inadvertently absolve healthcare professionals of responsibility or liability. This may involve developing explicit guidelines for the application of the method, establishing robust monitoring mechanisms, and promoting transparency and accountability in the design and implementation of the system.

In conclusion, the proposed method contains great potential for reducing clinician workload and enhancing the effectiveness of medical diagnosis. However, additional research is required to investigate its adaptability to diverse scenarios, optimize its performance for different use cases, and integrate it into a larger framework for decision support. In order to ensure its successful adoption in clinical practice, it is also essential to address the human factors and ethical considerations associated with its implementation.

## 8. Conclusions and Outlook

The research presented in this paper demonstrates that automatic classification of data derived from ICM devices can be applied effectively for various practical applications with the pipeline presented, demonstrating superior performance compared to existing state-of-the-art techniques, making this a milestone in the efforts to automate ICM data analysis.

The results of our experiments demonstrate that our method outperforms current methods (better F1 value) in most heart rhythm categories (the exception being normal heart rhythm, for which the sensitivity of our method is lower, but the specificity is significantly higher, which is arguably what we want for normal heart rhythms) for ICM data.

As such, our study demonstrates that despite inherent complexities, achieving a satisfactory level of automatic diagnosis with ICM device data is feasible. This includes the use of ECG clustering as a supplementary tool for medical professionals managing increasing volumes of ECG episodes and the implementation of clustering as a preparatory step for segment-adaptive parameter optimization during post-processing. Notably, our proposed method excels in dealing with underrepresented categories in ICM sECG data sets.

As unlabeled ICM data rapidly become one of the most prevalent types of ECG data, its anticipated volumetric growth emphasizes the importance of precise labeling. Unfortunately, the lack of specific labels diminishes its practical utility. This study reveals that contemporary classification algorithms struggle with such data sets. Nonetheless, the method proposed in this study provides a promising foundation for annotating such data, which could pave the way for large ECG episode databases with labels. This could circumvent existing restrictions and unlock a variety of applications.

In addition, the algorithms used in the presented pipeline, namely hard 10 s episode segmentation, CEEMD-based high-frequency noise detection, Lorenz-plot-based data transformation, and clustering, are all useful for the analysis of ECG episodes and could each be used individually depending on the use case.

The present study pioneers a flexible methodology that enables users to develop customized extensions and modifications. Several possibilities within the proposed pipeline have already been investigated, but future research may consider additional refinements to optimize overall performance based on the data and devices in use. Some of these are as follows. (i) Label prediction could be limited to high-confidence sub-episodes (as indicated by low *p*-values), which may be an effective strategy for evaluating automated classification while leaving more complex sub-episodes for manual diagnosis. (ii) As an alternative to static 10 s sub-episodes, dynamic episode segmentation may be utilized. This strategy would entail the identification of specific instances during an episode in which the cardiac rhythm shifts, enabling the segmentation of episodes based on this criterion. This would result in sub-episodes with similar cardiac rhythm types and therefore, potentially, a training set that can be learned and clustered more accurately and yield higher prediction accuracy (refer to Section 2.1 for details regarding inaccurate labels). (iii) In addition to the high-frequency-noise detector presented in this study, non-high-frequency-noise-detection techniques, such as clipping or missing data, should be considered for comprehensive noise reduction and thus further increase the model's accuracy by, e.g., not wrongly classifying normal rhythm episodes as low heart rate due to clipping (which affects QRS complex detection) or a part of the sub-episode being missing, thus leading to a low count of R peaks. (iv) Alternatives to Lorenz-plot transformations (e.g., morphology related features) could be used. (v) A systemic way of finding optimal parameters for DBSCAN clustering of the transformed sub-episodes could be developed.

In conclusion, this study provides not only a powerful new tool for the automatic classification of ICM device data but also a foundation upon which further advancements can be built.

**Author Contributions:** Conceptualization, A.B., A.L., B.J. and T.O.F.C.; methodology, A.B. and T.O.F.C.; validation, A.B., A.L., B.J., T.O.F.C. and B.H.D.; formal analysis, A.B.; investigation, A.B.; data curation, A.L. and B.J.; writing-original draft preparation, A.B.; writing-review, A.L., B.H.D. and T.O.F.C.; writing—editing, A.B. and T.O.F.C.; supervision, T.O.F.C.; project administration, T.O.F.C., A.L. and B.H.D. All authors have read and agreed to the published version of the manuscript.

**Funding:** This research was funded by the German Ministry of Research and Education (BMBF) project grants 3FO18501 (Forschungscampus MODAL), 01IS18025A and 01IS18037I (Berlin Institute for the Foundations of Learning and Data—BIFOLD).

**Institutional Review Board Statement:** Ethical review and approval were waived due to the complete anonymization of the ECG data, ensuring no identifiable information was present. One of the primary ethical concerns in research involving human subjects is the protection of their identity and personal information. In our study, the ECG data were fully anonymized, meaning all identifiable markers, attributes, or any other information that could potentially link the data to a specific individual were meticulously removed before the data were used in this study.

**Informed Consent Statement:** Patient consent was waived due to the complete anonymization of the ECG data, ensuring no identifiable information was present.

**Data Availability Statement:** The data presented in this study are available on request from the corresponding author.

**Conflicts of Interest:** A.L., B.J. and B.H.D. are employed by BIOTRONIK SE & Co., KG. The authors declare no other conflict of interest. The funders had no role in the design of the study; in the collection, analyses, or interpretation of data; in the writing of the manuscript; or in the decision to publish the results.

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
