# Peer review of "Enhancing Electrocardiogram (ECG) Analysis of Implantable Cardiac Monitor Data: An Efficient Pipeline for Multi-Label Classification"

_make, doi:10.3390/make5040077_

Round 1

Reviewer 1 Report

Dear authors,

Overall, your research shows promise and makes valuable contributions to the field. 

Here are some comments and questions I have:

1.While you mention the challenges related to labeling unlabeled ICM data, it would be beneficial to provide more details on the specific issues faced and potential solutions or strategies to address them.

2.When discussing the flexibility of the proposed methodology for customization, consider providing examples or case studies to illustrate how users can leverage this flexibility effectively.

3.The idea of dynamic episode segmentation based on cardiac rhythm shifts is intriguing. Please elaborate on how this approach would work in practice and its potential benefits.

4. When mentioning non-high-frequency noise detection techniques, such as clipping or missing data, provide insights into how these techniques could be integrated into your methodology for more comprehensive noise reduction.

5.While you touch on future research possibilities, consider expanding on specific areas that might benefit from additional investigation and refinement within the proposed pipeline.

Author Response

Thank you very much for your review. Please see our comments and revised paragraphs in the attachment under Reviewer 1

Reviewer 2 Report

1. Explain the novelty of the work. Mention the aim, methodology, and results in abstract sections.

2. In the Introduction stress more on the scope, baseline information, and the problem is connected to the paper.

3. Add the objectives of the paper in points. Add organization of the paper after the introduction.

4. A more recent literature survey is required, and compare with them. Add some recent work.

5. Conclusion to be made more systematic and highlight more on the results and add future scope of the paper.

6. The below papers have some interesting implications that you could discuss in your introduction and how they relate to your work.

doi.org/10.1007/s10916-019-1497-9

DOI : 10.5121/ijaia.2011.2204 

It can be improved.

Author Response

Thank you very much for your review. Please see our comments and revised paragraphs in the attachment under Reviewer 2

Reviewer 3 Report

The work presented herein addresses the design of an algorithm to automatically process ECG data from implantable cardiac monitors that provide for faster and accurate classification. The presented results how that it outperforms methods based on the Baseline Feature based and Baseline deep-learning based approaches.

The rational behind the need for developing this approach is very well presented. The limitations of other methods are highlighted, and the scenario in which the method is applicable is identified.

It should be clarified whether the proposed method is meant to be run in a clinic server after the data is transmitted, or if it can also be implemented within the capturing implantable cardiac monitor.

The title of section 4 should be changed to a more specific title and to avoid misunderstanding with the previous comment in (see section 5.1 for more details).

It is not clear what the D attached to the numbers in “(approximately 7600D for an episode or 1300D for a sub-episode)” mean. Also, it is not adequate to start section 6 with “The following section” when it refers to section 6 itself.

The legend of figure 7 should clarify to what clusters do the various colors correspond to.

Author Response

Thank you very much for your review. Please see our comments and revised paragraphs in the attachment under Reviewer 3
